GABRP promotes CD44s-mediated gemcitabine resistance in pancreatic cancer

Chen Chen 1
Wu Binfeng 1
Wang Mingge 1
Chen Jinghua 1
Huang Zhaohui 2
Shi Jin-Song shijs@jiangnan.edu.cn 1
1 Key Laboratory of Carbohydrate Chemistry and Biotechnology, Ministry of Education, School of Life Sciences and Health Engineering, Jiangnan University , Wuxi , Jiangsu , China
2 Wuxi Cancer Institute, Affiliated Hospital of Jiangnan University , Wuxi , Jiangsu , China
Verghese Shilpi
Electronic publication date: 2022 Jul 11
Publication date: 2022
Volume: 10
Electronic Location ID: e12728
Received 2021 Jun 18; Accepted 2021 Dec 10
Copyright: ©2022 Chen et al.
Copyright year: 2022
Copyright holder: Chen et al.
License: This is an open access article distributed under the terms of the Creative Commons Attribution License, which permits unrestricted use, distribution, reproduction and adaptation in any medium and for any purpose provided that it is properly attributed. For attribution, the original author(s), title, publication source (PeerJ) and either DOI or URL of the article must be cited.
License URL: https://creativecommons.org/licenses/by/4.0/

Keywords: CD44, GABRP, Gemcitabine, Chemoresistance, GEO, Pancreatic cancer

Funding: The Natural Science Foundation of China, a Jiangsu-Young Investigator Award BK20190593 Jiangnan University Young Investigator Award (2050205) JUSRP11958 China Postdoctoral Science Foundation 2020M681492 Jiangsu Higher Education Institutions, China PPZY2015B146 This work was supported by grants from the Natural Science Foundation of China, a Jiangsu-Young Investigator Award (grant no. BK20190593 to Chen Chen), a Jiangnan University Young Investigator Award (2050205) (grant no. JUSRP11958 to Chen Chen), China Postdoctoral Science Foundation (grant no. 2020M681492 to Chen Chen) and the Top-notch Academic Programs Project of Jiangsu Higher Education Institutions, China (PPZY2015B146 to Jinghua Chen). The funders had no role in study design, data collection and analysis, decision to publish, or preparation of the manuscript.

==============================
Background

Pancreatic ductal adenocarcinoma (PDAC) has the worst five-year overall survival rate among all cancer types. Acquired chemoresistance is considered one of the main reasons for this dismal prognosis, and the mechanism of chemoresistance is unknown.

Methods

We previously identified a subpopulation of chemoresistant CD44high-expressing PDAC cells. Subsequently, we selected the candidate gene, gamma-aminobutyric acid receptor subunit Pi (GABRP), from three Gene Expression Omnibus datasets as the potential CD44 downstream target mediating the gemcitabine resistance. Loss and gain of function such as stable knockdown of CD44 by small hairpin (sh) RNA-mediated silencing technique and overexpression (O/E) of CD44s had been studied for comparing the gemcitabine resistance among CD44high-expressing cells, shCD44 cells, CD44low-expressing cells and O/E CD44s expressing cells. Functional assays including cell viability, colony formation, invasion, quantitative PCR and western blotting techniques were performed to validate the roles of CD44 and GABRP playing in mediating the gemcitabine resistance in pancreatic cancer cells.

Results

CD44s depletion significantly reduced gemcitabine resistance in shCD44 single clone cells compared to CD44high-expressing cells. Knockdown of CD44 cells formed less colonies, became less invasive and remarkably decreased the mRNA level of GABRP. While overexpression of CD44s had the opposite effect on gemcitabine resistance, colony formation and invasive property. Of note, long term gemcitabine resistant pancreatic cancer cells detected increased expression of CD44 and GABRP. Clinically, GABRP expression was significantly upregulated in the tissues of patients with pancreatic cancer compared to the normal samples, and the overall survival rate of patients with low GABRP expression was longer. CD44 and GABRP co-expression was positively correlated in 178 pancreatic cancer patients.

Conclusion

Our findings suggest that GABRP may serve as a CD44s downstream target to diminish gemcitabine resistance in pancreatic cancer, and both CD44s and GABRP molecules have the potential to become prognostic biomarkers for PDAC patients with gemcitabine resistance.

Introduction

Pancreatic ductal adenocarcinoma (PDAC) is considered the most aggressive cancer among all cancer types, and the five-year survival rate of pancreatic cancer is less than 8% in Western countries. It is predicted to become the second leading cause of cancer-related deaths by 2030 (Rahib et al., 2014). The existing screening biomarker, CA19-9, cannot accurately diagnose pancreatic cancer, therefore, more candidate biomarkers need to be identified. Although modified 5-fluorouracil/leucovorin with irinotecan and oxaliplatin (FOLFIRINOX), and gemcitabine/nab-paclitaxel treatments significantly improve the survival of advanced PDAC patients, this comes at the expense of additional toxic side effects (Conroy et al., 2018; Kamisawa et al., 2016). Furthermore, these treatments have a limited effect on the long-term survival rate and quality life of patients with PDAC. This is partially due to the development of chemoresistance, and precision medicine or targeted therapy is urgently needed. We initiated the data mining of Gene Expression Omnibus (GEO) datasets focusing on gemcitabine-resistant versus sensitive cells, and pancreatic cancer verses adjacent non-cancerous tissues, and identified the overlapping gene, gamma-aminobutyric acid type A receptor subunit Pi (GABRP). GABRP primarily mediates inhibitory synaptic transmission in the mature central nervous system. It is also expressed at low levels in the uterus, ovaries, prostate, and breast (Wali et al., 2019), but is found at relative higher level in pancreatic cancers (Takehara et al., 2007). In vitro, transient knockdown of GABRP attenuated PDAC cell viability, while gamma-aminobutyric acid binding to GABRP promoted PDAC cell proliferation (Takehara et al., 2007). Upregulation of GABRP significantly contributed to pancreatic tumor growth and metastasis (Jiang et al., 2019). However, the functional role of GABRP-mediated chemoresistance in any cancer type has not been explored, especially in pancreatic cancer. Cancer stem cells (CSCs), characterized by CD24-, CD44-, and EpCAM-positivity in pancreatic cells, play important roles in metastasis and chemoresistance (Gogolok et al., 2020). CD44, a non-kinase receptor, is a CSC marker in many cancer types. Our previous study showed that PDAC cells expressing a high level of CD44 possessed a more mesenchymal phenotype and were gemcitabine-resistant, which gave rise to rapid tumor growth, while CD44low-expressing cells were more responsive to gemcitabine in a PDAC xenograft nude mouse model (Zhao et al., 2016). Abrogation of Snail or Twist, two key transcription factors involved in the epithelial-to-mesenchymal transition, enhanced gemcitabine sensitivity and increased the overall survival rate of genetically modified mice (Zheng et al., 2015). Other studies confirmed our findings that CD44-positive drug resistant cancer stem-like cells isolated from pancreatic cancer cells formed larger colonies and spheroids (Ling et al., 2018). These results indicate that suppression of CD44-positive/mesenchymal phenotype/chemoresistant PDAC cells may lead to better therapeutic outcomes.

The first four and last five exons of CD44 are constitutively expressed, and are known as the standard form of CD44 (CD44s), while other CD44 variants can be generated by alternative splicing in humans (Chen et al., 2018). The functions of various CD44 isoforms in pancreatic cancer are not fully understood. Downstream molecular targets of CD44 isoforms that mediate pancreatic cancer chemoresistance need to be identified. We hypothesized that GABRP is required for CD44 isoforms to induce chemoresistance in pancreatic cancer. A loss-of-function study indicated that stable knockdown of CD44s significantly reduced gemcitabine resistance compared to CD44s high-expressing cells. A GEO analysis identified the candidate gene, GABRP. Silencing of CD44s decreased GABRP expression at the transcriptional level. These findings suggest that GABRP may serve as a CD44s downstream target to affect CD44s-mediated chemoresistance. Both CD44 and GABRP were highly expressed in 178 pancreatic cancer patients which were validated in the starbase database. The current study provides new perspectives for improving the efficacy of gemcitabine in pancreatic cancer patients.

Methods

GEO data mining

GEO (https://www.ncbi.nlm.nih.gov/gds) provides a platform for data collection, processing, and normalization. We performed integrated analyses of GEO datasets of our interest (GSE15471 (Badea et al., 2008), GSE28735 (Zhang et al., 2012) and GSE36563 (Van den Broeck et al., 2012)) that contained matching samples either between normal and pancreatic tumor samples or parental and gemcitabine-resistant tumor cells (Idichi et al., 2017; Zhang et al., 2013). Differentially expressed genes (DEGs) of drug resistance versus non-drug treatment crossing the tumor or normal samples were overlapped by a Venn diagram. A p-value < 0.01, and an absolute value of —logFC (fold change) — > 2 were considered as the cut-off criteria.

Cancer cell line encyclopedia database analysis

The Cancer Cell Line Encyclopedia from the Broad Institute of MIT and Harvard (https://portals.broadinstitute.org/ccle) is a public resource containing gene expression and sequencing information of nearly 1,000 human cancer cell lines (Barretina et al., 2012). We used this resource to analyze the distribution of GABRP mRNA expression in cell lines derived from different tumor types.

Overall survival analysis

The five-year survival rate was determined by a Kaplan–Meier analysis between GABRP high- and low-expression groups using 178 pancreatic cancer patient samples and 4 normal samples. The log-rank test was performed to determine survival differences between the two groups. Differences were considered statistically significant at p < 0.05. The starBase website (http://starbase.sysu.edu.cn/) offered differential gene expression and survival analyses across 32 cancer types that were integrated from The Cancer Genome Atlas (Li et al., 2013).

Cell culture

CFPAC1-CD44high- and CD44low-expressing cells were initially separated by flow cytometry (Zhao et al., 2016). Single clones for CD44high expression, and stable knockdown CD44 cells, were generated in CFPAC1 cells. We used lentiviral vector-mediated transfection methods as described in our previous publication (Zhao et al., 2016). The cell lines were grown in RPMI 1640 medium supplemented with 10% fetal bovine serum (Gibco, Thermo Fisher Scientific, USA) and a 1% antibiotic mixture (100 U/mL penicillin and 100 µg/mL streptomycin) (Biotech, China). The cells were maintained in an incubator at 37 °C under 5% CO2.

Cell viability assay

Cells were seeded in triplicate in 96-well culture plates at a density of 3.0 × 103 cells per well, and treated with gemcitabine at different concentrations for 3 days. Cells were then incubated with MTT (Life Technologies, Thermo Fisher Scientific, China) solution (10 µL/well) and cultured at 37 °C for 3 h. Absorbance of each well was measured at 450 nm using an automatic microplate reader (Biofil, China). The cell viability percentage was calculated using the following equation: (absorbance of drug-treated sample/absorbance of untreated sample) × 100. All the experiments were repeated at least three times.

Quantitative real-time polymerase chain reaction

Total RNA was extracted from all cell lines using TRIzol (Life Technologies, Carlsbad, CA, USA) according to the manufacturer’s protocols. The RNA concentration was determined using a Nanodrop spectrometer (SMA4000 UV-VIS spectrophotometer, Merinton, China). One microgram of high-quality RNA was reverse-transcribed into cDNA (CWBio, Jiangsu, China). Then, 50 ng of cDNA were used to perform the quantitative real-time polymerase chain reaction using the SYBR Green kit (ABI Systems, Mississauga, ON, Canada). Samples were prepared in triplicate and normalized to β-actin using the 2ΔΔCt method. The results represent means ± standard deviation. Primer sequences of GABRP and ß-actin genes were list as below: GABRP forward primer: 5′-GCCCTAACAGAGCCTCAACA-3′; Reverse primer: 5′-TTGTCACTTCTGCCGACCTC-3′. ß-actin forward primer: 5′-GACCAATCCTGTCACCTC-3′;

Reverse primer: 5′-GATCTCCGTTCCCATTAAGAG-3′.

Transfection of pancreatic cancer cells

shRNA against CD44 were designed by the company (Genecopoeia, Rockville, MD, USA). The plasmid coated with Fugene 6 lipofectin, transfected to 293t cells, accompanied with lentiviral vectors (gift from Dr. Senlin Li’s lab, University of Texas Health Science Center, San Antonio, TX, USA). After packaging the live virus, the supernatant from 293t cells was harvested and applied to CD44high-expressing cells. Successful transfected cells were subsequently selected by puromycin for at least two weeks since the shRNA vector contained puromycin resistant gene as the selection marker. Likewise, the overexpression of CD44 vector was transfected using adenovirus and selected by puromycin resistant antibiotics for another 14 days. These cells were validated by qPCR experiments and western blotting analysis.

Colony formation assay

CD44high-expressing pancreatic cancer cells and two knockdown of CD44 single clone cells were counted and plated at the density of 500 cells per well. After 14 days, the cells were fixed by methanol and stained by 0.1% crystal violet. The cell numbers were counted under the microscope and pictured in prism software. Three independent experiments were conducted to calculate the standard deviation.

Invasion assay

Silence of CD44 (shCD44) in CD44high-expressing cells and its counterpart CD44high-expressing cells were seeded at the density of 5 × 104 cells in the upper chamber of the insert (Corning, USA) with the regular fresh RPMI medium (500 ul/insert) while the bottom chamber was fulfilled with the complete RPMI medium containing 10% FBS (700 ul/well) for 24 h. Then, the insert was washed by warm 1xPBS twice. And the cells were fixed by 80% ethanol for 15 min followed by 0.1% crystal violet staining for another 20 min. Fresh tap water was used to clean the stained insert and the well. At the end, the insert was wiped using the cotton swap to dry. The invasive cells were counted under the inverted microscope. Each group has three replicates and the experiments were repeated at least three times.

Western blot analyses

CD44high-expressing cells and two short hairpin (sh)RNA-CD44 single clone cells were lysed using radioimmunoprecipitation assay buffer (Beyotime, Shanghai, China) supplemented with protease and phosphatase inhibitors (tablet mini EDTA-free, Roche, Basel, Switzerland) on ice for 30 min. After centrifugation at 10,000 g for 5 min, the lysates were centrifuged at 10,000 g at 4 °C for 5 min. The supernatant was collected and the cell debris was discarded. Subsequently, the total protein concentration was determined by the bicinchoninic acid assay (CWBio, Cambridge, MA). A total of 50 µg of protein from each sample were separated by 8% sodium dodecyl sulfate polyacrylamide gel electrophoresis and transferred onto polyvinylidene difluoride membranes (Bio-Rad, Hercules, CA, USA). Afterwards, the membranes were blocked with 0.5% bovine serum albumin for 1 h and incubated with primary antibodies against CD44, as well as β-actin as the loading control (1:2000; biodragon, China). Then, the membranes were washed three times with 1 ×, tris-buffered saline/0.1% Tween 20 and incubated with secondary antibodies conjugated with horseradish peroxidase (anti-mouse IgG, 1:2000; Beyotime, Shanghai, China) for 1 h at room temperature. The protein bands were visualized with chemiluminescence, and the images were captured on a visualization instrument (Bio-Rad, Hercules, CA, USA).

Statistical analyses

All data are presented as means ± standard deviation. Prism 7 software (GraphPad, San Diego, CA, USA) was applied to perform Student’s t-test andANOVA test. P < 0.05 was considered statistically significant.

Results

DEGs comparing GABRP between human pancreatic cancer and normal tissues

Overcoming chemoresistance in pancreatic cancer remains the top priority among all therapies. To identify potential functional molecules selected using bioinformatics tools, we chose three independent GEO datasets: GSE15471, which contained pairs of normal and tumor tissue samples (Badea et al., 2008; Idichi et al., 2017); GSE28735, which compared the microarray gene expression profiles of 45 matching pairs of pancreatic tumors and adjacent non-tumor tissues (Zhang et al., 2013; Zhang et al., 2012); and GSE36563, which identified a side population resistant to gemcitabine (T3 vs. T4 chemoresistance) in human PDAC. The results uncovered a chemoresistant and CSC-associated phenotype (Van den Broeck et al., 2012). The microarray data were retrieved from the GEO dataset provided by the National Center for Biotechnology Information website and were initially analyzed by GEO2R software. Log2 fold change (FC) > 2 and p values < 0.01 were set as the cutoff criteria for subsequent analysis. Interestingly, GABRP was still identified as the overlapping gene GABRP gene (Fig. 1A). Therefore, we selected GABRP as our candidate gene to perform the various functional assays related to chemoresistance in PDAC.

Figure 1 Screen of differential gene expression from GEO dataset.

GSE15471: Whole tissue gene expression study of pancreatic ductal adenocarcinoma (tumor vs. normal tissue) GSE28735: Microarray gene expression profiles of 45 matching pairs of pancreatic tumor and adjacent non-tumor tissues from 45 patients with pancreatic ductal adenocarcinoma (cancer vs. adjacent non-tumor tissue) GSE36563: Human pancreatic adenocarcinoma contains a side population resistant to gemcitabine (T3 vs. T4 chemoresistance). (A) Venn diagram of potential overlapping genes from the chemoresistant pancreatic cancer cells compared to the normal cells within three datasets when logFC is more than 2. (B) Overall numbers of genes from three GEO datasets. (C) Differential expression of GABRP between 178 pancreatic cancer and 4 normal human tissues from the starbase database. (D) Overall survival rate comparison between GABRP high and low expression group in pancreatic cancer patients.

To explore the clinical relevance of GABRP in pancreatic cancer patients, we took advantage of starBase V3.0 (http://starbase.sysu.edu.cn/) (Li et al., 2013) where 178 cancer and four normal samples were analyzed in pancreatic adenocarcinoma (PAAD). Expression data showed that GABRP expression increased in cancer patients compared to that in the normal samples presented by the box-whisker plot (Fig. 1C). Subsequently, the Kaplan–Meier overall survival curve revealed that PAAD patients with lower GABRP expression survived significantly longer than those with relatively higher expression of GABRP, with 89 patients in both groups. The hazard ratio was 1.49 and the log-rank p value was 0.056 (Fig. 1D). Collectively, the discriminatory ability of GABRP can be reflected on DEGs and the overall survival rate in PAAD. This indicated that the GABRP signature could be a prognostic biomarker for predicting the patients’ lifespan.

Figure 2 MRNA expression of GABRP in different organs and cancer cell lines.

(A, B) mRNA expression of GABRP among different organs and cancer cell lines from both affy and RNAseq data sequenced by the different platforms.

Functional role of GABRP in a pancreatic cancer cell lines model

To validate the previously mentioned clinical data in our cell line model, we first assessed the mRNA expression of GABRP in various organs and cancer cell lines using the Cancer Cell Line Encyclopedia: https://portals.broadinstitute.org/ccle/page?gene=GABRP. This analysis showed relatively higher mRNA levels of GABRP in the upper aerodigestive system, bile ducts, and pancreas (Fig. 2A). RNAseq data offered a comparison of GABRP among different cancer cell lines and organs. Notably, the upper aerodigestive system, bile ducts, and pancreas again showed relatively higher GABRP mRNA levels compared to other organs. This is similar to the results indicated by the Affymax data (Fig. 2B). Because both platforms indicated that the pancreas contained higher levels of GABRP, we took advantage of our pancreatic cancer CFPAC-1 cells to test whether GABRP played an indispensable role in pancreatic cancer progression. First, CFPAC-1-CD44high and CD44low expressing cells were separated out using flowcytometry which obtained from our previous published work (Zhao et al., 2016). Then, we established shRNA-CD44 knockdown cells as described in our previous publication (Zhao et al., 2016). Single clone (s.c.) cells of shCD44 had been generated using limited dilution over 2 months period time. To eliminate clonal variation, we selected two clones labeled cl.1 and cl.2 that are depicted by gray and blue lines, respectively (Figs. 3A & 3B). On one hand, silencing CD44 using shCD44 in CD44high-expressing cells endowed those cells with less sensitivity to gemcitabine while CD44high-expressing cells were gemcitabine resistant (Fig. 3A), gemcitabine inhibited shCD44 cell proliferation in a dose dependent manner which 10 ng/ml, 20 ng/ml, 30 ng/ml and 40 ng/ml had significant reduction effect on cell proliferation than those lower doses in CFPAC-1-shCD44 cells (Fig. 3A, *p < 0.05, ***p < 0.001,****p < 0.0001). On the other hand, two CD44s overexpression single clones were successfully created in CD44low-expressing cells (Zhao et al., 2016). Cell viability assay showed that overexpression of CD44s drove higher gemcitabine resistance than those CD44low-expressing cells (Fig. 3B, *p < 0.05, **p < 0.01, ***p < 0.001, ****p < 0.0001). Consistently, two CD44s overexpression cells promoted the gemcitabine resistance significantly in a dose dependent manner even at the small dose starting from 2.5 ng/ml to higher dose of 40ng/ml (Fig. 3B). Next, we performed a series of functional assays including colony formation assay, invasion assay, real-time quantitative PCR and western blotting techniques using two sets of cell line models: CD44high-expressing cells versus shCD44 counterpart knockdown cells and CD44low-expressing cells versus CD44s overexpression cells. Colony formation suggested two shRNA-CD44 single clone cells from CD44high-expressing cells decreased colonies critically than those CD44high-expressing cells (Fig. 3C, ***p < 0.001, ****p < 0.0001). The invasion assay displayed that both silence of CD44 single clone cells detected significantly less invasive numbers of cells than the CD44high-expressing cells (Fig. 3D). Western blot analysis confirmed that we successfully diminished CD44s standard isoforms in shRNA-CD44 s.c cells, with ß-actin serving as the loading control (Fig. 3E). Lastly, CFPAC1 cells (CF-ctrl) were treated with increased concentrations of gemcitabine until 1,000 ng/ml over two months to generate gemcitabine resistant cell line depicted as CF-GR. Interestingly, we found both CD44 and GABRP protein levels elevated in gemcitabine resistant cells compared to the control none drug treated cells (Fig. 3G). These results indicated that CD44 and GABRP expressions were positively correlated and may contributed to gemcitabine resistance in pancreatic cancer cells. Another possibility is GABRP may potentially serve as a downstream target of CD44. Therefore, we knockdown CD44 from CD44high-expressing cells to test whether GABRP expression changed upon CD44 inhibition. Within our expectation, the expression of GABRP mRNA was reduced dramatically in CFPAC-1- CD44 Hi -shCD44 cells compared to CD44 Hi cells (Fig. 3F). Collectively, CD44s was required for gemcitabine resistance because its knockdown in both single clone cells significantly decreased cell growth, and these cells were more drug sensitive compared to the control cells (Fig. 3A). GABRP may contribute to CD44-induced gemcitabine resistance in pancreatic cancer (Fig. 3F).

Figure 3 MRNA expression of GABRP and gemcitabine sensitivity comparison between CFPAC-1-CD44 Hi and stable knockdown of shRNA-CD44 PDAC cells.

(A) Cell viability assay to test drug sensitivity among CD44high cells and its two stable knockdown of CD44 single clone cells. (B) Cell viability assay to test drug sensitivity among CD44low cells and its two overexpression of CD44s single clone cells. (C) Colony formation comparison among CD44high cells and its two stable shCD44 single clone cells. (D) Invasion assay comparison among CD44high cells and its two silence of CD44 single clone cells. (E) CD44 protein levels comparison between CD44high single clone cells and the silence of CD44 single clone cells. (F) GABRP mRNA expression between CFPAC1-single clone CD44high and shRNA-CD44 PDAC cells. (G) Gemcitabine resistant cell line expressed higher level of CD44 and GABRP compared to gemcitabine sensitive control PDAC cells.

Clinical relevance of CD44 and GABRP in pancreatic cancer patients

To validate the clinical significance of CD44 and GABRP in patients with pancreatic cancer, a Spearman’s correlation analysis was conducted based on the starBase V3.0 website to analyze co-expression of CD44 and GABRP in various cancers (Fig. 4A). The top cancers with significant p-values are listed in the Fig. 4. The highest Spearman’s correlation coefficient R values of the top three cancer types were colon adenocarcinoma (R = 0.377), pancreatic adenocarcinoma (R = 0.369), and stomach adenocarcinoma (R = 0.314, Fig. 4A). All three of these cancer types are digestive tract cancers, suggesting digestive cancer tissues from patients tend to express both CD44 and GABRPmolecules. Using 178 pancreatic adenocarcinoma patient samples, CD44 and GABRP expressions were significantly co-expressed in the linear regression model. This suggests that CD44 may directly regulate GABRP expression to affect chemoresistance. However, further experiments are required to verify our hypothesis.

Figure 4 Co-expression of CD44 and GABRP in pancreatic cancer patients.

(A) CD44 and GABRP were highly correlated in several cancer types from the starbase website. (B) CD44 and GABRP expressions were significantly correlated in 178 pancreatic cancer samples from the Starbase website.

Discussion

Chemoresistance is a major problem in addition to advanced metastasis in pancreatic cancer. Gemcitabine is the first-line therapy for pancreatic cancer approved by the United States Food and Drug Administration in 1997 (Burris 3rd et al., 1997). However, despite minimal improvement after the treatment, recurrent disease and gemcitabine resistance remain urgent concerns that hinder survival outcomes. For example, patients using the most recent combinational regimens, FOLFIRINOX (Kamisawa et al., 2016) and modified-FOLFIRINOX, had significantly better disease-free survival rates than those treated with gemcitabine alone (21.6% vs. 12.8%) (Zeng et al., 2019). However, the toxicity of the drug combinations was high and unsuitable for the older people.

Our previous study showed that CD44s-positive pancreatic cancer cells were more mesenchymal and gemcitabine-resistant. A xenograft nude mouse model confirmed our in vitro chemoresistance assays (Zhao et al., 2016). Specifically, nude mice injected with CD44low-expressing cells receiving gemcitabine weekly stayed gemcitabine-sensitive for over 20 weeks, while those injected with CD44s High-expressing cells escaped confinement from gemcitabine around 13 weeks and the tumors grew rapidly from 13 to 20 weeks (Zhao et al., 2016). Targeting CD44s blocks pancreatic tumor formation and post-radiotherapy recurrence in patients (Li et al., 2014). The CD44v6 isoform peptide reduces metastasis of human orthotopic pancreatic tumors in nude mice (Matzke-Ogi et al., 2016). However, no single molecular targeted therapy in pancreatic cancer has been successful for patients due to tumor heterogeneity. Thus, it is necessary to develop additional diagnostic biomarkers of chemoresistance of pancreatic cancer to help decipher the heterogeneity in pancreatic cancer patients.

We took advantage of GEO datasets to identify GABRP as a candidate gene that is differentially expressed between gemcitabine-resistant pancreatic cancer and normal cells (Fig. 1). Other researchers using The Cancer Genome Atlas and Oncomine databases also found that GABRP was a differentially expressed candidate gene to identify potential diagnostic and therapeutic targets in PDAC (Chang et al., 2020). In colon adenocarcinoma, GABRP is associated with a prognostic factor (Yan et al., 2020). Hypomethylated high-expression genes, including GABRP identified from GEO datasets and a GEO2R online analysis, suggested it to be a potentially effective biomarker for nasopharyngeal carcinoma (Wu, Zhou & Sun, 2020). Therefore, we investigated the downstream targets of CD44s using a loss-of-function study in which shRNA against CD44 remarkably reduced GABRP mRNA levels, and increased gemcitabine sensitivity in PDAC cells compared to the parental CD44s High-expressing cells (Fig. 3).

GABRP may directly serve as a downstream target, or indirectly trigger other signaling pathways that affect gemcitabine resistance. GABRP enhances ovarian carcinoma cell metastasis through activation of mitogen-activated protein kinase/extracellular signal-regulated kinase (Sung et al., 2017). Our current data showed that depletion of CD44s in pancreatic cancer cells rendered them chemoresistant, and a significant reduction in GABRP was observed in CD44s knockdown cells (Fig. 3). This suggests that the extracellular signal-regulated kinase pathway may be affected downstream of GABRP in shRNA-CD44 chemoresistant cells.

c-Met was reported as a CD44 co-receptor (Delitto et al., 2014) that mediates its downstream signaling to affect cancer growth and metastasis. c-Met inhibitors reduced the population of CSCs and had a synergistic inhibitory effect with gemcitabine to reduce pancreatic tumor growth in a NOD SCID mouse model (Li et al., 2011). Vascular endothelial growth factor receptor 2 is also known as a CD44 co-receptor (Matzke-Ogi et al., 2016). CD44 promotes the epithelial-to-mesenchymal transition in pancreatic cancer via activation of Snail, which regulates the expression of membrane type 1 metalloprotease (Jiang et al., 2015). GABRP is a disseminated tumor cell marker in the metastatic breast cancer field (Lacroix, 2006). Our current study indicated that GABRP may be considered as a CD44 downstream target. Therefore, developing GABRP antibodies or inhibitors may have potential synergistic suppressive functions in gemcitabine-resistant PDAC cells.

Conclusion

We built the first link between GABRP and gemcitabine in pancreatic cancer, and we report, for the first time, that GABRP might be downstream of the CD44 effector gene in gemcitabine-resistant pancreatic cancer cells. The expression of both CD44 and GABRP was upregulated in pancreatic gemcitabine resistant cells and tumor tissues compared to the null resistant cells and normal tissues, and they were significantly co-expressed in patients with pancreatic cancer. Targeting CD44 and GABRP may have a synergistic effect on the suppression of gemcitabine-induced resistance in pancreatic cancer.

Supplemental Information

Supplemental Information 1 Cell line origin

Click here for additional data file.

Supplemental Information 2 QPCR raw data

Click here for additional data file.

We thank the CEO Lao Xinyuan and co-founder Tan Zhujun from Helixlife Company (https://www.helixlife.cn/) for providing the online courses to help us datamine the bioinformatics websites.

Additional Information and Declarations

Competing Interests

Author Contributions

Data Availability

The authors declare there are no competing interests.

Chen Chen conceived and designed the experiments, performed the experiments, analyzed the data, prepared figures and/or tables, and approved the final draft.

Binfeng Wu performed the experiments, prepared figures and/or tables, and approved the final draft.

Mingge Wang analyzed the data, prepared figures and/or tables, and approved the final draft.

Jinghua Chen analyzed the data, prepared figures and/or tables, and approved the final draft.

Zhaohui Huang analyzed the data, prepared figures and/or tables, and approved the final draft.

Jin-Song Shi conceived and designed the experiments, authored or reviewed drafts of the paper, and approved the final draft.

The following information was supplied regarding data availability:

The raw data are available as Supplemental Files.

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
