# Peer review of "GABRP promotes CD44s-mediated gemcitabine resistance in pancreatic cancer"

_PeerJ, doi:10.7717/peerj.12728_

## Round 0.1 · original submission · Major Revisions

Please address all the concerns of the reviewers.

Reviewer 1 ·

Basic reporting

No comment

Experimental design

No comment

Validity of the findings

No comment

Additional comments

Manuscript Number (#60030)
The manuscript is fairly well written and discusses how GABRP promotes CD44s-mediated gemcitabine resistance in pancreatic cancer. In this manuscript, the authors report the first link between GABRP and gemcitabine in pancreatic cancer.

Additional concern:
1. Although this finding reports the first time GABRP as a downstream of the CD44 effector gene in gemcitabine-resistant pancreatic cancer cells, detailed information on how GABRP promotes CD44s mediated gemcitabine resistance is missing in the manuscript.
2. The major issue is the use of a single cell line to validate the hypothesis tested. To validates the findings, at least two cell lines must be used to validate the hypothesis tested.
3. Mechanism of action involved in CD44 directed GABRP expression affecting gemcitabine resistance should be included in the manuscript.
4. Another major issue that needs to be addressed is the possibility of the non-specificity of shRNA. The knockdown data shown in the figures show the use of only include a single shRNA, and the Methods and Results do not mention how shRNA cells were made. Authors should use more than one shRNA for the study.
5. Authors should also include a western blot of GABRP for figure 3B.
6. Authors should show the GABRP protein expression in tumor tissues using immunohistochemistry or immunofluorescence or the pancreatic tissue microarray.
7. Authors should proofread the manuscript for spelling mistakes.

Reviewer 2 ·

Basic reporting

Introduction is written very broadly, revise to fit to scope of the manuscript.

Manuscript writing is not meeting the publication standards.

Experimental design

Could you rewrite methods section in detail. FBS can not be used for culturing stem cells (CD44 high) and there is no mentioning of other media in this section.

Remove the word 'single clone' throughout the manuscript. Use standard nomenclature for the knockdown cell line.

Is it appropriate to use CD44 knockdown cells from CD44 High instead of using CD44 High and CD44 low cells, explain.

Validity of the findings

The manuscript has novel concept of establishing the role of GABRP in chemoresistance of pancreatic cancer stem cells.

However due to very less evidence/data shown in the manuscript, it is highly recommended to do more functional assays to support the hypothesis.

Additional comments

In fig1c, mention the p values.
In fig3c, present the data from one CD44 sh clone. It is always recommended to use two shRNAs rather than multiple clones of single shRNA.
In fig3d, confirm GABRP levels by western blot also.
In fig3, "CF-cl.21-shCD44-cl.1" "CD44 High single cloneshCD44 single clone cells" Replace with standard nomenclature.

Reviewer 3 ·

Basic reporting

No Comment

Experimental design

Experimental design is not that much strong enough to support the hypothesis. For example Functional role of CD44 shRNA in spheroid formation or any other functional assays like proliferation, migration and colony formation.

Authors were not mentioned how many replicates they did.

Validity of the findings

No Comment

Additional comments

"GABRP promotes CD44s-mediated gemcitabine resistance in pancreatic cancer" by Chen chen group is quite interesting and very important in most of the cancers.
But the authors use most of the data from data sets (online), not from original experiments, whatever they use the data, that is good and enough to support the hypothesis, but I feel if it is more experimentally proven.
Here is the some comments to improve the manuscript.
1. Line 43 says some cancers, what are those cancers ?
2. Functional role of CD44 shRNA in spheroid formation or any other functional assays like proliferation, migration and colony formation.
3. Authors were not mentioned how many replicates they did.

---

## Round 0.2 · accepted · Accept

Thanks for addressing all concerns by the reviewers.

Reviewer 1 ·

Basic reporting

'no comment'

Experimental design

'no comment'

Validity of the findings

'no comment'

Additional comments

The authors have satisfactorily answered the raised concerns.

Reviewer 2 ·

Basic reporting

No comment

Experimental design

By carefully analyzing GEO datasets authors identified GABRP overlaps between datasets, found that high expression of GABRP is associated with poor patient survival and finally provided strong evidence that CD44 and GABRP expression are significantly correlates positively. And in figure 3 authors proved this concept using western blot and RTPCT where over expression of CD44 increased GABRP expression and knockdown of CD44 reduced GABRP expression. So far it is very good. However, authors have not shown any experimental evidence that shows CD44 expression is directly responsible for GABRP expression in PDAC cells. Because there is an equal probability of CD44 being directly involved in GABRP expression or it could be indirect regulation. without having solid evidence authors can not claim "GABRP promotes CD44s-mediated gemcitabine resistance in pancreatic cancer"

Provide experimental evidence how GABRP is involved in Gemcitabine resistance.

In fig 3 include CFPAC1 wild type cells as the reference in all the experiments.
Fig3 provide images for invasion and colony formation assay.
Fig 3e provide westernblot for GABRP

Fig 3 Authors used only one cell line, reproduce using additional PDAC cell lines

Validity of the findings

No comment